# Prognostic Impact of the AML60+ Score for Elderly Patients with Acute Myeloid Leukemia Treated with Hypomethylating Agents: A Retrospective Multicentric Analysis

**DOI:** 10.3390/cancers17162658

**Published:** 2025-08-14

**Authors:** Verena Petermichl, Stefan Fuchs, Matthias Weber, Katrin Gobat, Charlotte Micheloud, Lukas Graf, Yannick Gerth, Jeroen S. Goede, Thomas Lehmann, Christoph Driessen, Ulrich J. M. Mey, Richard Cathomas, Sergio Cogliatti, Tobias Silzle

**Affiliations:** 1Clinic for Medical Oncology and Hematology, Cantonal Hospital St. Gallen, 9007 St. Gallen, Switzerland; 2Department of Internal Medicine, Cantonal Hospital Graubünden, 7000 Chur, Switzerland; 3Institute of Pathology, Cantonal Hospital St. Gallen, 9007 St. Gallen, Switzerland; 4Swiss Cancer Institute, Statistics Unit, 3008 Bern, Switzerland; 5Center for Laboratory Medicine, St. Gallen, 9007 St. Gallen, Switzerland; 6Medical Oncology and Hematology, Cantonal Hospital Winterthur, 8400 Winterthur, Switzerland; 7Department of Medical Oncology and Hematology, Cantonal Hospital Graubünden, 7000 Chur, Switzerland

**Keywords:** acute myeloid leukemia, prognostication, hypomethylating agents, venetoclax, ELN2022, mPRS, AML60+

## Abstract

The treatment of acute myeloid leukemia (AML) in elderly patients has changed significantly with the introduction of venetoclax in combination with hypomethylating agents (HMA). The AML60+ score was developed to identify elderly patients who would benefit from intensive chemotherapy and allogeneic stem cell transplantation. The aim of our retrospective and multicenter study was to determine whether the AML60+ is also helpful in assessing the prognosis of elderly AML patients receiving HMA-based therapy. Patients with lower risk according to AML60+ lived significantly longer than those within the higher risk categories. The C-index, as a comparative measure of prognostic significance, was higher for AML60+ than for the two indices used for comparison, the molecular Prognostic Scoring System and the European Leukemia Net 2022 classification. Given the small sample size, analyses of larger cohorts are needed to confirm our observation.

## 1. Introduction

Acute myeloid leukemia (AML) and AML/Myelodysplastic neoplasms (AML/MDS) are heterogeneous clonal disorders of myeloid progenitor cells [1]. With a median age at diagnosis of about 70 years [2], they are primarily diseases of the elderly. In this patient group, comorbidities and frailty often lead to ineligibility for intensive treatments such as induction chemotherapy or allogeneic stem-cell transplantation.

Instead, these patients receive less intensive regimens. A current standard of care is the combination of hypomethylating agents (HMAs), such as azacitidine (Aza) or decitabine (Dec), with the BCL2 inhibitor venetoclax (Ven) [3,4]. While these treatments may not be as effective in the real world as they were in the trials that led to their approval, they provide a clinically meaningful benefit to this difficult-to-treat population [5,6]. Despite a significant risk of tumor lysis syndrome and infection in the early phase of treatment and myelosuppression later, HMA-based regimens can be safely used even in octogenarians and nonagenarians [7] when proper precautions are taken [8]. Long-term responses appear increasingly possible [9,10], even after stopping treatment [11,12].

The European Leukemia Net (ELN) 2022 classification [3] was developed using data from patients treated with intensive chemotherapy and proved to be of less value when applied to patients treated with HMA ± Ven. Therefore, several prognostic scores have been proposed for non-intensively treated patients. Some of them are based on the mutational profile alone, such as the molecular Prognostic Scoring System (mPRS) [13,14] or the “Genetic Risk Classification for Adults with AML Receiving Less Intensive Treatments” proposed by the ELN in 2024 (ELN2024) [15] while other systems still consider the cytogenetic profile in terms of adverse cytogenetic risk together with the mutational profile, such as the BEAT AML2024 or the MAYO risk model [16,17].

In contrast, the AML60+ score considers age and sex as patient-related factors and the white blood cell count together with cytogenetics (monosomal karyotype according to ELN2022) and mutation profile (*TP53*, *FLT3-*ITD, *ASXL1*, *DNMT3A,* and *RUNX1*) as disease-related factors [18]. It is designed to identify elderly patients who may benefit from intensive treatment, including conventional induction and allogeneic stem cell transplantation. Its prognostic value in the context of less intensive regimens, including HMA +/− Ven, is currently unknown.

Therefore, we aimed to evaluate its prognostic significance in the context of HMA-based therapies in comparison to the ELN2022 classification and the mPRS [13,14], as this score was developed specifically for patients on HMA + Ven combinations.

## 2. Materials and Methods

### 2.1. Study Cohort

Patients aged 60 years or older were included in this retrospective and multicentric chart review, if they were diagnosed with either acute myeloid leukemia according to the 2022 International Consensus Classification (ICC2022) [19] or the 2022 World Health Organization (WHO) classification [20], or with MDS/AML according to ICC2022, at one of the participating centers (Cantonal Hospital Graubünden, Cantonal Hospital St. Gallen and Cantonal Hospital Winterthur, all in Switzerland) between 2017 and 2024. Patients were eligible if they were ineligible for intensive treatment (including induction chemotherapy with either the conventional “7 + 3” regimen or CPX-351 and allogeneic stem cell transplantation) in the opinion of the treating physician and if they had received at least one cycle of an HMA-based regimen as first-line treatment. Patients with acute promyelocytic leukemia and with relapsed/refractory disease were excluded. All cases were individually reviewed prior to inclusion in the dataset to ensure correct classification.

We retrospectively collected clinical and laboratory data from medical records as documented at the time of diagnosis (±30 days) and before the start of treatment, including cytogenetic and molecular data.

### 2.2. Molecular Profiling

For patients whose diagnostic work-up did not include next-generation sequencing (NGS), molecular profiling was performed retrospectively if DNA was available from the respective samples obtained at diagnosis, as described in the Appendix A.

### 2.3. Prognostic Scoring Systems

Patients were classified according to ELN2022 [3] and according to mPRS [13,14]. The AML60+ score was calculated as described by Versluis et al. [18]. Components and risk groups of the AML60+ score are shown in the Appendix A.

### 2.4. Statistical Analysis

Categorical variables were described using frequencies and percentages and compared using the χ^2^ test. Continuous variables were described by median and interquartile range (IQR) or range, and compared using the Mann–Whitney U test. Overall survival was calculated in months from the date of diagnosis to the respective event date, i.e., death or censoring. Survival functions were estimated using the Kaplan–Meier approach and survival curves were compared using the log-rank (if not otherwise specified) or Breslow test. For pairwise comparisons, the Benjamini–Hochberg procedure was applied to the *p*-values to correct for multiple testing. Median survival times with 95% confidence intervals (CIs) are reported. Hazard ratios (HRs) with 95% CIs were estimated using univariable Cox proportional hazards (PH) regression models. Continuous explanatory variables were log-transformed prior to inclusion in the Cox models. The PH assumption was assessed visually and, when appropriate, using Schoenfeld residuals. In addition, the concordance probability estimate (C-index) was calculated for the Cox models as a metric of model discrimination. All analyses were performed using R Statistical Software (v4.5; R Core Team, R Foundation for Statistical Computing, Vienna, Austria) or IPSS (Version 25.0, IBM Corp., Armonk, NY, USA).

## 3. Results

### 3.1. Patient Population

A total of 142 patients were identified (female n = 68 [47.9%], male n = 74 [52.1%]). Median age was 77 years (range 61–90). A total of 122 patients (86%) were diagnosed with AML and 20 (14%) with MDS/AML. Detailed patient characteristics, including the distribution of disease entities according to ICC2022, are shown in Table 1.

During follow-up (median 8 months, range 0–68; database closure on 4 June 2025), 114 patients (80%) died (25/114 [22%] within 30 days of diagnosis) and 1 patient was lost to follow-up.

The majority of patients received a combination of an HMA and Ven (Aza and Ven n = 80 [56.3%], Dec and Ven n = 16 [11.3%]). A total of 32 patients (22.5%) received Aza and 11 patients (7.7%) received Dec alone. A total of 3 patients (2.1%) received Dec and ibrutinib in the HOVON135/SAKK30/15 trial [21]. No patient received an *IDH1/2* inhibitor as part of the first-line treatment.

Complete information on cytogenetic alterations (conventional metaphase cytogenetics or comparative genomic hybridization combined with a fluorescence in situ hybridization panel allowing detection of translocations according to ELN2022) was available for 122 patients (86%). NGS data were available for 129 patients (89.4%), including 6 patients for whom NGS was performed retrospectively.

Overall, the available data allowed risk stratification according to mPRS in 121 patients (85.2%) and according to ELN2022 in 117 patients (82.4%). The AML60+ score was available for 105 patients (73.9%). All three scores were available for 102 patients (71.8%).

### 3.2. Distribution of Risk Groups According to the Different Scores

As shown in Figure 1A, according to ELN2022, 10/117 (8.5%) patients were classified as favorable, 16/117 (13.7%) as intermediate, and 91/117 (77.8%) as adverse.

A monosomal karyotype was present in 28 patients (23.9%) and a complex karyotype in 11 patients (7.7%). The majority of patients with a monosomal or complex karyotype (28/39, 72%) had a *TP53* mutation (VAF > 1%). A *TP53* mutation without complex or monosomal karyotype was found in 6/117 patients (4.2%). Two patients with a complex karyotype and six patients with a monosomal karyotype were *TP53* wild-type. In three patients with a monosomal or complex karyotype, *TP53*-status was unknown. A detailed overview of the individual frequencies of the ELN2022 classification factors can be found in the Appendix A.

A total of 121 patients could be stratified according to the mPRS. The distribution of the risk groups according to mPRS (higher benefit n = 68 [56.2%], intermediate benefit n = 18 [14.9%], and lower benefit n = 35 [28.9%]) is shown in Figure 1B and the distribution of risk factors is shown in detail in Appendix A.

The AML60+ was available for 105 patients (73.9%). A total of 33 patients (31.4%) were classified as very poor risk, 36 (34.3%) as poor risk, and 34 (32.4%) as intermediate risk. Only two patients (1.9%) were classified as favorable (see Figure 1C). Therefore, favorable and intermediate risk patients were analyzed together for survival analyses. The distribution of the individual risk factors in the single risk groups according to AML60+ is shown in the Appendix A.

### 3.3. Risk Stratification of Patients Stratified According ELN2022 by AML60+

Both AML60+ and ELN2022 were available for 102 patients. The majority (7/9, 77.8%) of patients stratified as favorable by ELN2022 had an intermediate risk profile according to AML60+. The remaining two patients (22.2%) were classified as poor risk. The intermediate risk group according to ELN2022 (n = 15) consisted of two patients (13.3%) with favorable risk, eight patients (53.3%) with intermediate risk, four patients (26.7%) with poor risk, and one patient with very poor risk according to AML60+. The ELN2022 adverse risk group (n = 78) consisted of 17 (21.8%) intermediate risk patients, 29 (37.2%) poor risk patients, and 32 very poor risk patients according to AML60+. See Figure 2.

### 3.4. Risk Stratification of Patients Stratified According ELN2022 by mPRS

Both mPRS and ELN2022 were available for 108 patients. Within the favorable risk group, according to ELN2022 (n = 9), eight patients (88.9%) were classified as higher benefit according to mPRS and the remaining patient was classified as intermediate benefit. Of the 15 ELN2022 intermediate risk patients, 10 patients (66.7%) were classified as higher benefit and 5 (33.3%) as intermediate benefit. The adverse risk category according to ELN2022 (n = 84) was divided into 40 patients with higher benefit (47.6%), 9 patients with intermediate benefit (10.7%), and 35 patients with lower benefit (41.7%). See Figure 2.

### 3.5. Prognostic Impact of the Single Risk Factors According to the AML60+ in Univariate Analysis

As shown in Figure 3, both a monosomal karyotype and a TP53 mutation were associated with a significantly shorter median OS (Figure 3D,H).

Mutations in *ASXL1*, *FLT3-ITD*, or *RUNX1* were also associated with a shorter median OS, but the difference was not significant (Figure 3E,G,I). No statistically significant association was observed between *DNMT3A* mutations and survival (Figure 3F).

Male patients (n = 74, 52.1%) had a significantly shorter OS than female patients (median 5 months [95% CI 3–7] versus 15 months [95% CI 10–19], *p* = 0.034; Figure 3B). For age as a continuous variable, there was no evidence of an association with OS after log-transformation of age (HR 0.88; 95% CI 0.06–13.8, *p* > 0.9). However, the Schoenfeld test indicated a violation of the proportional hazards assumption (see Appendix A), so this result should be interpreted with caution. The OS of patients aged 65 years or older was shorter than that of patients aged <65 years, but the difference was not significant (Figure 3A). Given the small sample size, we did not attempt to identify an age-related cut-off that might be more appropriate for prognostication.

For the log-transformed white blood cell (WBC) count, Cox regression analysis provided strong evidence of an association with OS (HR 1.18, 95% CI 1.05–1.34, *p* = 0.008), without evidence of a violation of the proportional hazards assumption (Appendix A). Correspondingly, a white blood cell count > 20 × 10^9^/L was associated with a shorter median overall survival (10 months [95%CI 6–14] versus 5 months [95%CI 2–10], *p* = 0.130 [log rank] and *p* = 0.046 [Breslow]; see Figure 3C).

Patients with *K/N-RAS*- or *FLT3*-ITD mutations had higher WBCs at diagnosis (median 9.6 × 10^9^/L [IQR 2–43.7] versus 2.8 × 10^9^/L [IQR 1.69–12.0], *p* = 0.026) and patients with a WBC > 20 × 10^9^/L were more frequent within *K/N-RAS*- or *FLT3*-ITD mutated cases (no *K/N-RAS*—or *FLT3*-ITD mutation: 20/104 [19%] versus 8/19 [42%], *p* = 0.034).

### 3.6. Frequency and Prognostic Impact of NPM1 and IDH2 Mutations in the Risk Groups According to AML60+

An *IDH2* mutation without negative prognostic factors according to mPRS (*FLT3*-ITD, *NRAS*, *KRAS,* or *TP53* mutations) was present in 15/105 patients (14%). *IDH2* mutations were significantly more frequent in patients with favorable/intermediate risk than in those with poor/very poor risk (10/36 [28%] versus 5/69 [7%], *p* = 0.007) and were associated with a favorable OS (median 19 months [95% CI 14.5-NR] versus 7 months [95% CI 5–13], *p* = 0.038; see Appendix A).

*NPM1* mutations without *NRAS*, *KRAS*, *TP53,* or *FLT3*-ITD mutations were present in 9/105 patients (9%) and were significantly more frequent in the intermediate/favorable risk group than in the poor/very poor risk groups (7/36, [19%] versus 2/69 [3%], *p* = 0.007). These patients had a longer median OS (19 months [95% CI 15–23] versus 9 months [95% CI 1.5-NR]). However, this difference was not statistically significant (*p* = 0.150). See Appendix A.

Taken together, nearly 50% of patients in the favorable/intermediate category according to AML60+ (17/36, [47%]) had an *IDH2* or *NPM1* mutation without an additional molecular risk factor according to mPRS.

### 3.7. Prognosis According to ELN2022, mPRS, and AML60+

The overall survival of the individual risk groups according to ELN2022, mPRS, and AML60+ for all patients is shown in Figure 4 and for patients treated with HMA + Ven in Figure 5. For each score, there is evidence that at least one of the individual risk groups has a statistically different survival curve.

However, for the overall cohort, pairwise comparisons of Kaplan–Meier estimates using the log-rank test, and corrected for multiple testing, showed a significant difference in OS only between intermediate and adverse risk according to ELN2022 (corrected log-rank *p* = 0.021), but not between favorable and adverse (corrected log-rank *p* = 0.062) or favorable and intermediate (corrected log-rank *p* = 0.95).

For mPRS, OS was significantly different between higher and lower benefit (corrected log-rank *p* = 0.001), but not between lower and intermediate benefit (corrected log-rank *p* = 0.449) or intermediate and higher benefit (corrected log-rank *p* = 0.192).

OS by AML60+ was significantly different between all risk groups: favorable/intermediate risk versus poor (corrected log-rank *p* < 0.001) or versus very poor (corrected log-rank *p* < 0.001) and between poor and very poor (corrected log-rank *p* = 0.007).

When only patients treated with HMA + Ven were analyzed (see Figure 5), no difference was observed between the individual risk groups according to ELN2022 (favorable vs. adverse: log-rank *p* = 0.069; favorable vs. intermediate: *p* = 0.45 and intermediate vs. adverse: *p* = 0.069; all *p*-values corrected for multiple testing).

OS difference, according to mPRS, was significant between higher and lower benefit (corrected log-rank *p* < 0.001), but not between lower and intermediate benefit (corrected log-rank *p* = 0.397) or between intermediate and higher benefit (corrected log-rank *p* = 0.11).

Again, OS was significantly different between all risk groups according to AML60+: intermediate/favorable risk versus poor: corrected log-rank *p* = 0.023; intermediate/favorable versus very poor risk: corrected log-rank *p* < 0.001; and poor versus very poor risk corrected log-rank *p* = 0.005.

An analysis of patients treated with HMA without Ven revealed a significant survival difference only for the risk groups according to AML60+ (intermediate/favorable versus poor: corrected log-rank *p* = 0.02 and versus very poor: corrected log-rank *p* = 0.02), but not for the risk groups according to ELN2022 or mPRS (see Appendix A for further details).

Univariable Cox regression models with, for each score, the highest risk category as reference category revealed the lowest hazard ratio for AML60+ intermediate/favorable patients compared to very poor risk (all patients: HR 0.17 [95% CI 0.10, 0.31], *p* < 0.001; HMA + Ven treated patients: HR 0.14 [95% CI 0.07, 0.30], *p* < 0.001; HMA without Ven HR 0.25 [95% CI 0.09–0.70], *p* = 0.008).

Detailed results of the Cox regression analyses for all three scores are shown in Table 2 (all patients), in Table 3 (patients treated with HMA + Ven), and in the Appendix A for patients treated with HMA without Ven.

Because the sample sizes are small and the proportional hazards assumption appears to be violated for ELN2022 and mPRS (see Figure 4A,C and Figure 5A,C) these results must be interpreted with caution.

### 3.8. Comparison of the C-Indices

The concordance score was calculated for patients for whom all three scores were available (patients treated with HMA alone or HMA + Ven, n = 102). It was highest for AML60+ (0.67), followed by mPRS (0.60) and ELN2022 (0.58). When only patients receiving HMA + Ven (n = 69) or HMA without Ven (n = 33) were analyzed, similar results were obtained (HMA + Ven: concordance score AML60+ 0.69, mPRS 0.64, ELN 2022 0.60; HMA without Ven: concordance score AML60+ 0.62, mPRS 0.56, and ELN2022 0.54).

## 4. Discussion

Our analysis of elderly AML patients treated with HMA with or without Ven suggests that the AML60+ score may be a valuable prognostic tool for this population, although it was originally developed to identify elderly patients who would benefit from intensive treatment, including allogeneic stem cell transplantation [18].

The combined favorable/intermediate group according to AML60+ showed a median overall survival of 40 months for patients treated with HMA + Ven and, as shown by the C-indices, the AML60+ was superior to both ELN2022 and mPRS. Therefore, the AML60+ may help to identify a group of patients with a favorable prognosis or even long-term responders that are missed by the ELN2022 classification and not as clearly captured by the mPRS.

*NPM1* and *IDH2* mutations are among the mutations most associated with favorable outcomes to treatment with HMA +/− Ven, when they occur in the absence of signaling mutations and together with *TP53* wild-type (for review [22,23]). In our cohort, both constellations are significantly enriched in the intermediate/favorable group according to AML60+. This implies that the AML60+ score indirectly captures a substantial proportion of patients with a favorable molecular risk profile without specifically testing for the respective mutations.

On the other hand, approximately 50% of patients in the intermediate/favorable group of AML60+ did not have *NPM1* or *IDH2* mutations. Thus, the unique combination of patient-related factors (age and sex), with conventional disease-related factors (WBC), and oncogenomic markers provided by the AML60+ identifies additional patients who can expect a favorable outcome following HMA-based treatment.

Regarding sex, a significantly better response to HMA has been reported in female patients with high-risk MDS [24], and the inclusion of sex added prognostic information to all standard prognostic tools for MDS [25]. In addition, male patients with high-risk MDS or AML showed an inferior outcome after treatment with Dec in the prospective HOVON135 study [26]. With regard to HMA + Ven treatment, male sex was an independent negative prognostic marker for overall and relapse-free survival in the MAYO prognostic score cohort [17], and female patients were overrepresented in the favorable risk group according to BEAT-AML2024 [16]. In addition, sex-related differences in the prognostic impact of co-occurrence patterns have been described for selected mutations [27]. On the other hand, no negative prognostic effect of sex was observed in a National Health Service (NHS) analysis of 587 patients treated with HMA + Ven [28]. Despite the latter observation, the consideration of sex may be one point that explains the better performance of AML60+ in our cohort. Recently, it has been suggested to include sex as a prognostic marker in AML prognostication [29] and our observations imply that this is useful in the context of HMA-based treatment as well.

Another advantage of the AML60 + may be the inclusion of age as a second patient-related factor. According to real-world data from the NHS on the use of HMA + Ven, increasing age proved to be an independent risk factor independent of AML subtype (secondary and/or therapy-related AML) and oncogenomic profile. [28] In secondary AML, age added prognostic information independent of the mutational profile in patients treated with HMA + Ven. [30]. Advanced age has also shown to be an independent negative prognostic marker, independent of mutational profile, in a Japanese AML cohort treated with Ven-based regimens [31], and was also prognostic in a large series of elderly AML patients from the MAYO Clinic [32].

AML patients with an elevated WBC count are at higher risk for complications such as tumor lysis syndrome, coagulopathies, or infections, especially in the early phase of treatment [33]. In patients treated with HMA and low-dose cytosine arabinoside, a higher WBC count was associated with a shorter OS in an analysis of the PETHEMA group [34] and a negative prognostic impact of WBC > 10 × 10^9^/L was described in a Chinese cohort [35]. A higher WBC is often associated with *FLT3*-ITD or *K/N-RAS* mutations [36,37], as it was the case in our cohort. Therefore, a WBC > 20 × 10^9^/L may additionally represent a surrogate marker for the presence of RAS-mutations, which are not directly considered by the AML60+.

Furthermore, it may be relevant that according to AML60+, both *TP53* mutations and a monosomal karyotype are considered as individual risk factors, each with a high weight. *TP53* mutations are a marker of very poor prognosis with low response rates and dismal survival [38,39] even with the combination of HMA + Ven [40]. They show a high association with a complex or monosomal karyotype [41,42] and the majority of patients with the latter abnormalities are identified by the presence of a *TP53* mutation. However, some patients with a monosomal or complex karyotype have a *TP53* wild-type. A pooled analysis of two pivotal trials of HMA + Ven suggested that in this situation, the adverse prognostic impact of poor-risk cytogenetics may not apply when they are treated with Aza + Ven [43].

However, in an analysis of a prospective phase II trial of Dec + Ven [44], a complex karyotype and especially monosomies of chromosomes 5, 7, and 17 were associated with worse survival. In another analysis of 301 patients treated with HMA + Ven, an unfavorable karyotype, according to ELN2022, remained an independent negative prognostic marker for both the achievement of a complete remission and for overall survival, independent of the *TP53* mutation status [45]. A complex karyotype was associated with reduced overall survival according to real-world data from the NHS [28]. Therefore, identifying patients with an unsatisfactory response to HMA-based therapies is likely to be aided by considering both *TP53* mutations and adverse karyotypes.

In addition to *FLT3*-ITD and *TP53* mutations, mutations in *ASXL1*, *DNMT3A,* and *RUNX1* are risk factors according to AML60+. Their prognostic role in the context of HMA-based treatments is not as well defined and they are not specifically considered by other currently proposed prognostic scoring systems. AML patients with mutated *ASXL1* showed high response rates to HMA + Ven, but with lower rates of MRD-negativity and, therefore, higher risk of relapse [45]. Accordingly, *ASXL1* mutations represent an independent negative prognostic marker in an analysis of the NHS cohort [28]. In case of relapse after HMA-based treatment, *ASXL1* mutations are associated with a worse prognosis independent of mutations in *TP53* and *RAS* [46].

*DNMT3A* mutations were associated with better response rates and longer overall survival in the MAYO cohort, but did not retain independent prognostic value in multivariable models [17]. In an analysis of Chinese AML patients with myelodysplasia-related changes receiving low-intensity Ven-based treatments, mutated *DNMT3A* was associated with a higher risk of relapse [47]. When occurring as a co-mutation in *IDH*-mutated cases, mutated *DNMT3A* is likely to confer an unfavorable prognosis [48]. This finding may be particularly relevant because *IDH2*-R172 mutations often co-occur with mutated *DNMT3A* [49]. A similar negative prognostic effect of *DNMT3A* mutations co-occurring with *NPM1* mutations has been suggested in the context of various induction therapies, including Ven-based low-intensity treatments [50].

*RUNX1* mutations were found with a high frequency of approximately 40% in primary Ven-refractory cases and were mutually exclusive with *TP53* mutations [51]. When present as the only adverse factor according to ELN2022, they were associated with lower OS when intensive chemotherapy or HMA were used, but not with HMA + Ven as treatment [52]. Within the MAYO cohort [17], *RUNX1* mutations were associated with lower response rates and shorter survival in both univariate and multivariate analyses, but were not included in the final model with censoring at the time of allogeneic stem cell transplantation.

Taken together, these data suggest that the mutational status with respect to *ASXL1, DNMT3A,* and *RUNX1* may add prognostic value at least in combination with the other variables in the AML60+ score.

The main limitations of our study primarily include its retrospective nature and the limited sample size, which precluded assessing the prognostic impact of most of the individual AML60+ factors or identifying alternative cut-offs for age and WBC as continuous variables. As a retrospective study based on data collected during clinical routine, not all parameters necessary to calculate the individual scores were available for all patients, implying a risk of selection bias. In addition, the treatments were heterogeneous, and no patient received a combination of Ven with an *IDH1/2* inhibitor or low-dose cytosine arabinoside. However, all key observations were confirmed in patients treated with HMA + Ven, which is the current standard of care. Neither *DDX41* nor *MLL* were covered by the NGS panels used in the routine diagnostic laboratories collaborating with the participating centers. Therefore, we were not able to compare the AML60+ score with the ELN2024 classification [15] or the BEAT-AML 2024 score [16], which provide alternative options for prognostication in daily clinical practice.

## 5. Conclusions

Our data show for the first time that the AML60+ can be used as a prognostic tool in elderly patients treated with HMA alone or in combination with Ven. The AML60+ may even be more accurate than the mPRS, which was developed specifically for patients treated with HMA + Ven. However, further studies analyzing larger datasets, ideally derived from prospective registry data, are needed to confirm our observations. In addition, it is likely that some modifications of the AML60+ would improve its prognostic accuracy, such as defining an age cut-off that is more appropriate for a target population that often includes octogenarians or considering adverse cytogenetics according to ELN2022 instead of only a monosomal karyotype. Due to the small sample size, we were not able to answer these questions. Despite its limitations, our analysis underscores that conventional metaphase cytogenetics and simple patient- and disease-related factors such as WBC, sex, and age can help refine the prognosis in AML apart from the mutational profile.

## Figures and Tables

**Figure 1 cancers-17-02658-f001:**
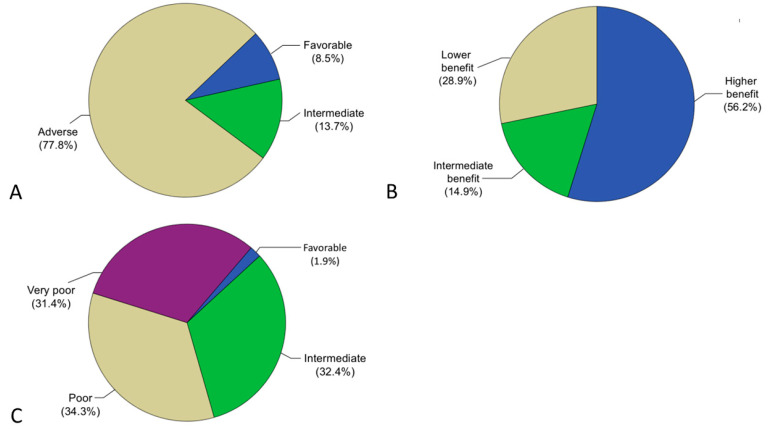
Distribution of the risk groups according to ELN2022 (**A**), mPRS (**B**), and AML60+ (**C**).

**Figure 2 cancers-17-02658-f002:**
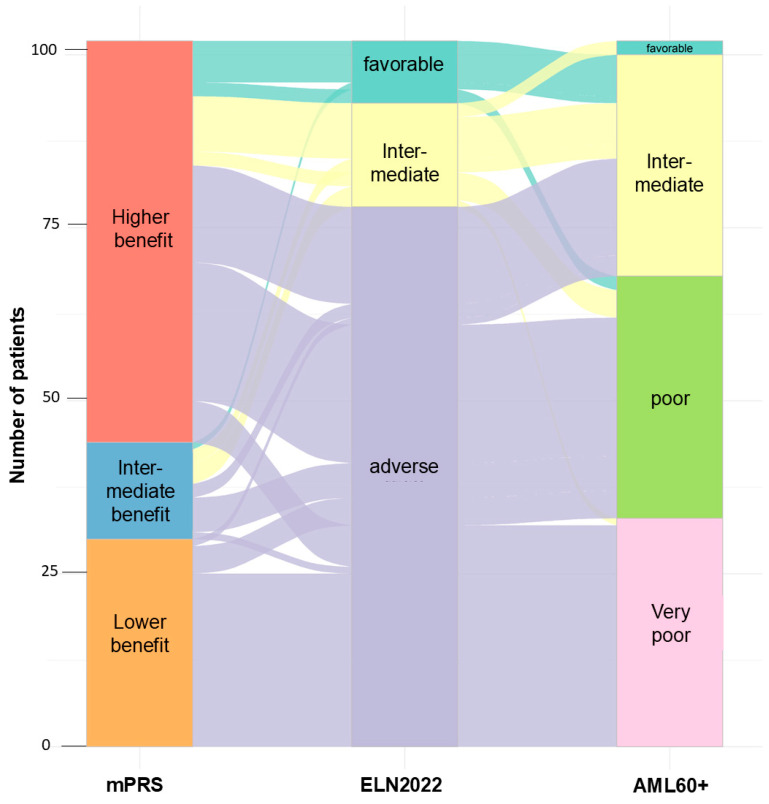
River plot of patients for whom all three scores were available (n = 102).

**Figure 3 cancers-17-02658-f003:**
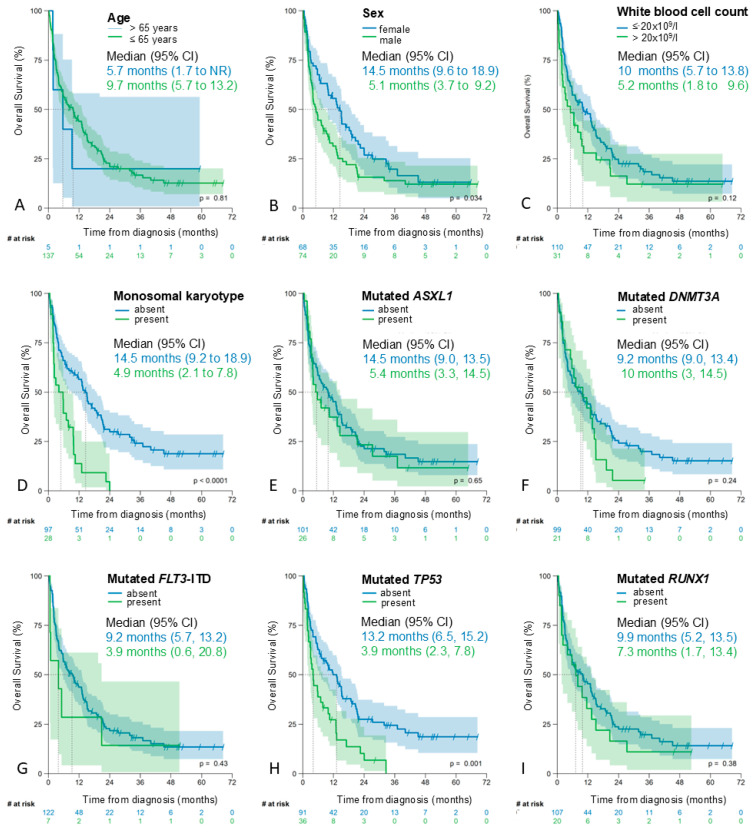
Kaplan–Meier estimates of overall survival by the single factors according to AML60+. (**A**) age, (**B**) sex, (**C**) white blood cell count, (**D**) monosomal karyotype, (**E**) mutated *ASXL1*, (**F**) mutated *DNMT3A*, (**G**) *FLT3*-ITD mutation, (**H**) mutated *TP53* and (**I**) mutated *RUNX1*.

**Figure 4 cancers-17-02658-f004:**
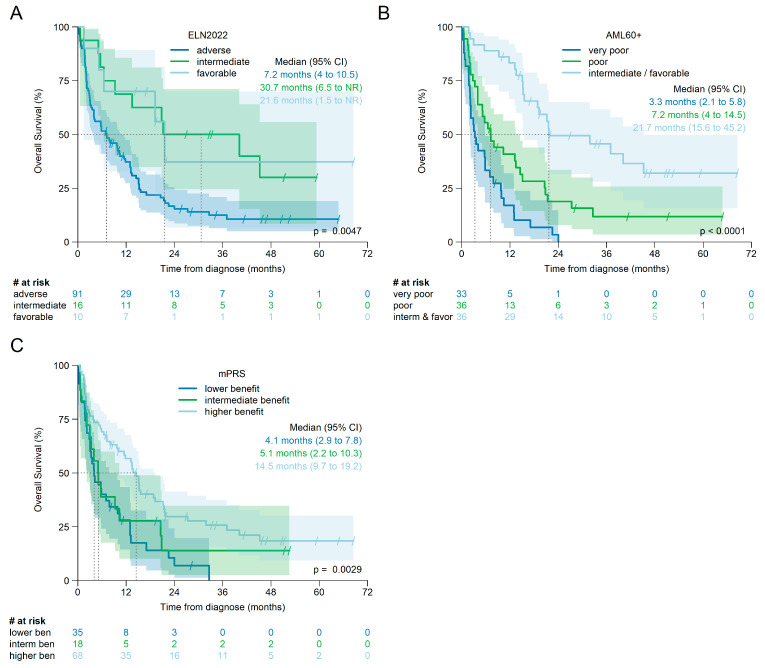
Kaplan–Meier estimates of overall survival for patients treated with either HMA or HMA + Ven by ELN2022, n = 117 (**A**), AML60+, n = 105 (**B**), and mPRS, n = 121 (**C**).

**Figure 5 cancers-17-02658-f005:**
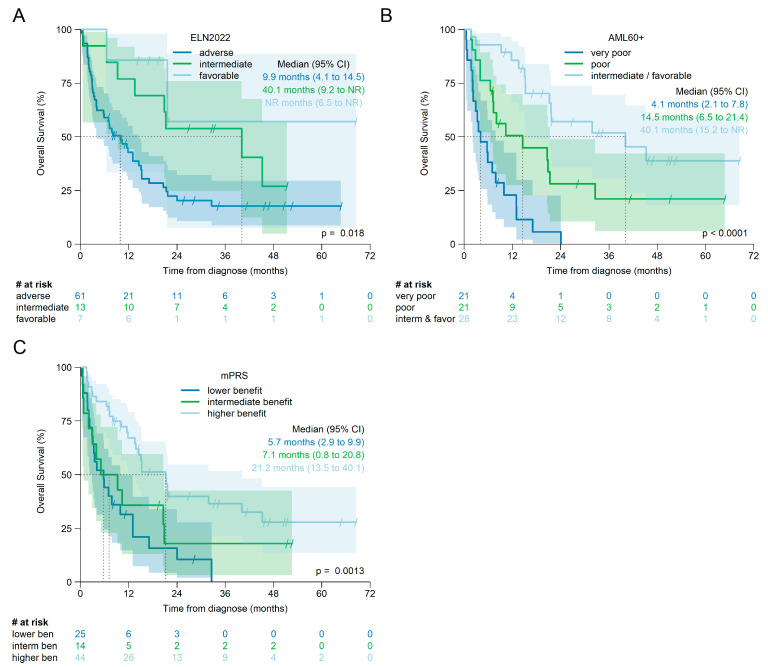
Kaplan–Meier estimates for overall survival for patients treated with HMA + Ven by ELN2022, n = 81 (**A**), AML60+, n = 70 (**B**), and mPRS, n = 83 (**C**).

**Table 1 cancers-17-02658-t001:** Patient characteristics.

Characteristic	Whole Population
n	142
Age [years], median, (IQR)	77 (74–81)
Female n, (%)	68 (48)
Male n, (%)	74 (52)
AML according to ICC2022 n, (%)	122 * (86)
AML with recurrent genetic abnormality n, (%)	21 * (14)
*AML with* *Mutated NPM1*	*17 (11.5)*
*AML with * *KMT2A * *Rearrangement*	*3 (2)*
* AML with inv(3)(q21.3q26.2) *	*1 (0.5)*
AML with mutated *TP53* n, (%)	26 (18)
AML with myelodysplasia-related gene mutation n, (%)	45 (32)
AML with myelodysplasia-related cytogenetic abnormality n, (%)	4 (2.8)
AML not otherwise specified n, (%)	13 (9)
Subtype unknown due to insufficient work-up n, (%)	13 (9)
MDS/AML according to ICC2022 n, (%)	20 (14)
MDS/AML with mutated *TP53* n, (%)	8 (5.6)
MDS/AML with myelodysplasia-related gene mutation n, (%)	9 (6.3)
MDS/AML not otherwise specified n, (%)	1 (0.7)
Subtype unknown due to insufficient work-up n, (%)	2 (1.4)
Hemoglobin	
available	141/142
[g/L], median (IQR)	88 (76–102)
Leukocytes	
available	141/142
(×10^9^/L), median (IQR)	3.3 (1.8–15)
Platelet count	
available	141/142
(×10^9^/L), median (IQR)	66 (36–111)
Neutrophils	
available	139/142
(×10^9^/L), median (IQR)	0.93 (0.36–3.1)
Blasts peripheral blood	
available	140/142
(%), median, IQR	8.3 (0.5–23)
Blasts bone marrow	
available	141/142
(%) median, IQR	35 (25–63)

* Including one patient with 6% bone marrow blasts and NPM1 mutation (AML according to WHO2022) IQR, Inter Quartile Range.

**Table 2 cancers-17-02658-t002:** Univariable Cox regression models for OS by ELN2022, mPRS, and AML60+ for patients treated with either HMA or HMA + Ven.

	HR	95% CI	*p*
**ELN2022**			
Adverse *	--	--	
Intermediate	0.41	0.21, 0.79	0.008
Favorable	0.40	0.16, 0.99	0.047
**mPRS**			
Lower benefit *	--	--	
Intermediate benefit	0.73	0.39, 1.36	0.3
Higher benefit	0.46	0.29, 0.73	<0.001
**AML60+**			
Very poor *	--	--	--
Poor	0.47	0.28, 0.78	0.004
Intermediate/favorable	0.17	0.10, 0.31	<0.001

* Reference category; Abbreviations: CI = Confidence Interval, and HR = Hazard Ratio.

**Table 3 cancers-17-02658-t003:** Univariable Cox regression models for OS by ELN2022, mPRS, and AML60+ for patients treated with HMA + Ven only.

	HR	95% CI	*p*
**ELN2022**			
Adverse *	--	--	
Intermediate	0.47	0.22, 1.00	0.049
Favorable	0.24	0.06, 0.97	0.045
**mPRS**			
Lower benefit *	--	--	
Intermediate benefit	0.67	0.32, 1.40	0.3
Higher benefit	0.36	0.20, 0.64	<0.001
**AML60+**			
Very poor *	--	--	--
Poor	0.32	0.16, 0.65	0.002
Intermediate/favorable	0.14	0.07, 0.30	<0.001

* Reference category; Abbreviations: CI = Confidence Interval, and HR = Hazard Ratio.

## Data Availability

The data presented in this study are available on request from the corresponding author.

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
