# Peer review of "Prognostic Impact of the AML60+ Score for Elderly Patients with Acute Myeloid Leukemia Treated with Hypomethylating Agents: A Retrospective Multicentric Analysis"

_cancers, 2025, doi:10.3390/cancers17162658_

Round 1
Reviewer 1 Report
Comments and Suggestions for Authors
This retrospective multicentric manuscript evaluated the prognostic utility of the AML60+ score in elderly patients with acute myeloid leukemia (AML) treated with hypomethylating agents (HMAs), with or without venetoclax (HMA-Ven). The study leverages a cohort of 142 patients from multiple Swiss hospitals, integrating next-generation sequencing (NGS) data and clinical characteristics to assess the AML60+ score’s performance compared to the European Leukemia Net 2022 (ELN2022) classification and the molecular Prognostic Scoring System (mPRS). The data suggested that the AML60+ score effectively stratifies patients into risk groups, with superior prognostic performance compared to ELN2022 and mPRS, particularly for those treated with HMA-Ven. While the manuscript is scientific interested, I have the following concerns, such as a relatively small sample size and incomplete statistical analyses, etc.
Here are major concerns:
- Small Sample Size: The cohort of 142 patients, while multicentric, is relatively small, limiting the statistical power and generalizability of the findings. Larger cohorts are critical for clinical adoption and a solid conclusion.
- Incomplete Statistical Details: The manuscript lacks clarity on certain statistical methods. For example, the adjusted log-rank test used for survival comparisons (page 11) does not specify which covariates were adjusted for. Similarly, the C-index comparison between AML60+, ELN2022, and mPRS lacks detailed reporting of confidence intervals or p-values to quantify statistical significance.
- Limited Discussion of Treatment Heterogeneity: The study includes patients treated with HMA alone or HMA-Ven, but the analysis does not fully explore how treatment differences impact the AML60+ score’s prognostic performance. A subgroup analysis is recommended to stratify outcomes by treatment type.
- Incomplete ELN2022 and mPRS Data: Supplementary Table S2 (Distribution of Risk Factors according to ELN2022) is incomplete in the provided document, making it difficult to evaluate the full context of ELN2022 risk stratification. Additionally, the mPRS stratification (Table S3) shows a high proportion of “other abnormalities” (56.5%), which lacks specificity and reduces interpretability.
- Lack of Mechanistic Insights: The manuscript does not explore why the AML60+ score outperforms ELN2022 and mPRS. For instance, the role of specific mutations (e.g., TP53, ASXL1) in driving prognostic differences is noted but not linked to underlying biological mechanisms, limiting the study’s contribution to understanding AML biology.
Here are minor concerns:
- Typographical and Formatting Errors: The manuscript contains typographical errors, such as “AML6+” instead of “AML60+” (Supplementary Figures S3, S4) and inconsistent gene nomenclature (e.g., “FLT5-TDD” instead of “FLT3-ITD” in Supplementary Figures S3, S4). Additionally, the repeated numbers (e.g., “272” on page 11, “46” on page 16) in the OCR text suggest potential formatting issues in the original document.
- Figure and Table Clarity: Supplementary Figures S1 and S2 (boxplots and Cox regression models for age and leukocyte count) are described but lack detailed legends explaining the axes or statistical outputs (e.g., hazard ratios, confidence intervals). Table 1 has inconsistent percentage calculations (e.g., “AML, with mutated TP3S3” at 8.5% for n=2 in a cohort of 142 is unclear).
- Acronym Consistency: Acronyms like “HIMA-Ven” (page 12) appear to be typos for “HMA-Ven.” Consistent use of acronyms and their definitions upon first use would improve readability.
- Data Availability: The manuscript does not address data availability, which is a concern for reproducibility.
- Patient Characteristics Discrepancy: Table 1 reports 141 patients (68 female, 73 male), but the Discussion mentions 105 patients. This discrepancy needs clarification to ensure consistency.
Author Response
Please see the uploaded Rebuttal letter

Reviewer 2 Report
Comments and Suggestions for Authors
The authors have conducted a timely and relevant study evaluating which of the currently available prognostic scoring systems best applies to a cohort of 142 AML patients treated with hypomethylating agents with or without venetoclax. This is an important and clinically meaningful question, as no validated prognostic score currently exists for this specific treatment setting.
The manuscript is well written, logically structured, and the rationale for the analyses is clearly and plausibly explained. Appropriate statistical methods were employed, including comparative evaluation of model performance. The application of the prognostic scoring systems appears accurate and thorough. The cited literature is appropriate and up to date. All figures are necessary, well chosen, and contribute meaningfully to the manuscript.
Minor comments:
In Supplementary Figure S1, the subpanels (A, B, and C) should be labeled.
In the legend of Supplementary Figure S3, "IDH1 status 1" likely was "IDH2 status 1" and should be corrected for clarity.
I recommend this manuscript for publication in Cancers after minor revisions.
Author Response
Please see the uploaded rebuttal letter
